# Soft Weight-Sharing for Neural Network Compression

**Karen Ullrich**
University of Amsterdam
`karen.ullrich@uva.nl`

**Edward Meeds**
University of Amsterdam
`tmeeds@gmail.com`

**Max Welling**
University of Amsterdam
Canadian Institute for Advanced Research (CIFAR)
`welling.max@gmail.com`

## Abstract

The success of deep learning in numerous application domains created the desire to run and train them on mobile devices. This however, conflicts with their computationally, memory and energy intense nature, leading to a growing interest in compression. Recent work by Han et al. (2015a) propose a pipeline that involves retraining, pruning and quantization of neural network weights, obtaining state-of-the-art compression rates. In this paper, we show that competitive compression rates can be achieved by using a version of "soft weight-sharing" (Nowlan & Hinton, 1992). Our method achieves both quantization and pruning in one simple (re-)training procedure. This point of view also exposes the relation between compression and the minimum description length (MDL) principle.

## 1 Introduction

"Bigger is better" is the ruling maxim in deep learning land. Deep neural nets with billions of parameters are no longer an exception. Networks of such size are unfortunately not practical for mobile, on-device applications which face strong limitations with respect to memory and energy consumption. Compressing neural networks could not only improve memory and energy consumption, but also lead to less network bandwidth, faster processing and better privacy. It has been shown that large networks are heavily over-parametrized and can be compressed by approximately two orders of magnitude without significant loss of accuracy. Apparently, over-parametrization is beneficial for optimization, but not necessary for accurate prediction. This observation has opened the door for a number of highly successful compression algorithms, which either train the network from scratch (Hinton et al., 2015; Iandola et al., 2016; Courbariaux & Bengio, 2016; Courbariaux et al., 2016) or apply compression post-optimization (Han et al., 2015b;a; Guo et al., 2016; Chen et al., 2015; Wen et al., 2016).

It has been long known that compression is directly related to (variational) Bayesian inference and the minimum description principle (Hinton & Van Camp, 1993). One can show that good compression can be achieved by encoding the parameters of a model using a good prior and specifying the parameters up to an uncertainty given, optimally, by the posterior distribution. An ingenious bits-back argument can then be used to get a refund for using these noisy weights. A number of papers have appeared that encode the weights of a neural network with limited precision (say 8 bits per weight), effectively cashing in on this "bits-back" argument (Gupta et al., 2015; Courbariaux et al., 2014; Venkatesh et al., 2016). Some authors go so far of arguing that even a single bit per weight can be used without much loss of accuracy (Courbariaux et al., 2015; Courbariaux & Bengio, 2016).

In this work we follow a different but related direction, namely to learn the prior that we use to encode the parameters. In Bayesian statistics this is known as empirical Bayes. To encourage compression of the weights to $K$ clusters, we fit a mixture of Gaussians prior model over the weights. This idea originates from the nineties, known as soft weight-sharing (Nowlan & Hinton, 1992) where it was used to regularize a neural network. Here our primary goal is network compression, but as was

shown in Hinton & Van Camp (1993) these two objectives are almost perfectly aligned. By fitting the mixture components alongside the weights, the weights tend to concentrate very tightly around a number of cluster components, while the cluster centers optimize themselves to give the network high predictive accuracy. Compression is achieved because we only need to encode $K$ cluster means (in full precision) in addition to the assignment of each weight to one of these $J$ values (using $\log(J)$ bits per weight). We find that competitive compression rates can be achieved by this simple idea.

## 2 MDL VIEW ON VARIATIONAL LEARNING

Model compression was first discussed in the context of information theory. The minimum description length (MDL) principle identifies the best hypothesis to be the one that best compresses the data. More specifically, it minimizes the cost to describe the model (complexity cost $\mathcal{L}^C$) and the misfit between model and data (error cost $\mathcal{L}^E$) (Rissanen, 1978; 1986). It has been shown that variational learning can be reinterpreted as an MDL problem (Wallace, 1990; Hinton & Van Camp, 1993; Honkela & Valpola, 2004; Graves, 2011). In particular, given data $\mathcal{D} = \left\{ \mathbf{X} = \{\mathbf{x}_n\}_{n=1}^N, \mathbf{T} = \{\mathbf{t}_n\}_{n=1}^N \right\}$, a set of parameters $\mathbf{w} = \{w_i\}_{i=1}^I$ that describes the model and an approximation $q(\mathbf{w})$ of the posterior $p(\mathbf{w}|\mathcal{D})$, the variational lower bound, also known as negative variational free energy, $\mathcal{L}(q(\mathbf{w}), \mathbf{w})$ can be decomposed in terms of error and complexity losses

$$\mathcal{L}(q(\mathbf{w}), \mathbf{w}) = -\mathbb{E}_{q(\mathbf{w})} \left[ \log \left( \frac{p(\mathcal{D}|\mathbf{w})p(\mathbf{w})}{q(\mathbf{w})} \right) \right] = \underbrace{\mathbb{E}_{q(\mathbf{w})} \left[ -\log p(\mathcal{D}|\mathbf{w}) \right]}_{\mathcal{L}^E} + \underbrace{\mathrm{KL}(q(\mathbf{w})||p(\mathbf{w}))}_{\mathcal{L}^C} \quad (1)$$

where $p(\mathbf{w})$ is the prior over $\mathbf{w}$ and $p(\mathcal{D}|\mathbf{w})$ is the model likelihood. According to Shannon's source coding theorem, $\mathcal{L}^E$ lower bounds the expected amount of information needed to communicate the targets $\mathbf{T}$, given the receiver knows the inputs $\mathbf{X}$ and the model $\mathbf{w}$. The functional form of the likelihood term is conditioned by the target distribution. For example, in case of regression the predictions of the model are assumed be normally distributed around the targets $\mathbf{T}$.

$$p(\mathcal{D}|\mathbf{w}) = p(\mathbf{T}|\mathbf{X}, \mathbf{w}) = \prod_{n=1}^N \mathcal{N}(\mathbf{t}_n|\mathbf{x}_n, \mathbf{w}) \quad (2)$$

where $\mathcal{N}(\mathbf{t}_n, \mathbf{x}_n, \mathbf{w})$ is a normal distribution. Another typical example is classification where the conditional distribution of targets given data is assumed to be Bernoulli distributed[1]. These assumptions eventually lead to the well known error functions, namely cross-entropy error and squared error for classification and regression, respectively.

Before however we can communicate the data we first seek to communicate the model. Similarly to $\mathcal{L}^E$, $\mathcal{L}^C$ is a lower bound for transmitting the model. More specifically, if sender and receiver agree on a prior, $\mathcal{L}^C$ is the expected cost of communicating the parameters $\mathbf{w}$. This cost is again twofold,

$$\mathrm{KL}(q(\mathbf{w})||p(\mathbf{w})) = \mathbb{E}_{q(\mathbf{w})} \left[ -\log p(\mathbf{w}) \right] - H(q(\mathbf{w})) \quad (3)$$

where $H(\cdot)$ denotes the entropy. In Wallace (1990) and Hinton & Van Camp (1993) it was shown that noisy encoding of the weights can be beneficial due to the bits-back argument if the uncertainty does not harm the error loss too much. The number of bits to get refunded by an uncertain weight distribution $q(\mathbf{w})$ is given by its entropy. Further, it can be shown that the optimal distribution for $q(\mathbf{w})$ is the Bayesian posterior distribution. While bits-back is proven to be an optimal coding scheme (Honkela & Valpola, 2004), it is often not practical in real world settings. A practical way to cash in on noisy weights (or bits-back) is to only encode a weight value up to a limited number of bits. To see this, assume a factorized variational posteriors $q(\mathbf{w}) = \prod q(w_i)$. Each posterior $q(w_i)$ is associated with a Dirac distribution up to machine precision, for example, a Gaussian distribution with variance $\sigma$, for small values of $\sigma$. This implies that we formally incur a very small refund per weight,

$$H(q(\mathbf{w})) = -\int_{\Omega} q(\mathbf{w}) \log q(\mathbf{w}) \, \mathrm{d}\mathbf{w} = -\int_{\mathbb{R}^I} \mathcal{N}(\mathbf{w}|\mathbf{0}, \sigma\mathbf{I}) \log \mathcal{N}(\mathbf{w}|\mathbf{0}, \sigma\mathbf{I}) = [\log(2\pi e \sigma^2)]^I. \quad (4)$$

---

[1]For more detailed discussion see Bishop (2006).

Note that the more coarse the quantization of weights the more compressible the model. The bits-back scheme makes three assumptions: (i) weights are being transmitted independently, (ii) weights are independent of each other (no mutual information), and (iii) the receiver knows the prior. Han et al. (2015a) show that one can successfully exploit (i) and (ii) by using a form of arithmetic coding (Witten et al., 1987). In particular, they employ range coding schemes such as the Sparse Matrix Format (discussed in Appendix A). This is beneficial because the weight distribution has low entropy. Note that the cost of transmitting the prior should be negligible. Thus a factorized prior with different parameters for each factor is not desirable.

The main objective of this work is to find a suitable prior for optimizing the cross-entropy between a delta posterior $q(\mathbf{w})$ and the prior $p(\mathbf{w})$ while at the same time keeping a practical coding scheme in mind. Recall that the cross entropy is a lower bound on the average number of bits required to encode the weights of the neural network (given infinite precision). Following Nowlan & Hinton (1992) we will model the prior $p(\mathbf{w})$ as a mixture of Gaussians,

$$p(\mathbf{w}) = \prod_{i=1}^{I} \sum_{j=0}^{J} \pi_j \mathcal{N}(w_i|\mu_j, \sigma_j^2). \tag{5}$$

We learn the mixture parameters $\mu_j$, $\sigma_j$, $\pi_j$ via maximum likelihood simultaneously with the network weights. This is equivalent to an empirical Bayes approach in Bayesian statistics. For state-of-the-art compression schemes pruning plays a major role. By enforcing an arbitrary "zero" component to have fixed $\mu_0 = 0$ location and $\pi_0$ to be close to 1, a desired weight pruning rate can be enforced. In this scenario $\pi_0$ may be fixed or trainable. In the latter case a Beta distribution as hyper-prior might be helpful. The approach naturally encourages quantization because in order to optimize the cross-entropy the weights will cluster tightly around the cluster means, while the cluster means themselves move to some optimal location driven by $\mathcal{L}^E$. The effect might even be so strong that it is beneficial to have a Gamma hyper-prior on the variances of the mixture components to prevent the components from collapsing. Furthermore, note that, mixture components merge when there is not enough pressure from the error loss to keep them separated because weights are attracted by means and means are attracted by weights hence means also attract each other. In that way the network learns how many quantization intervals are necessary. We demonstrate that behaviour in Figure 3.

## 3 RELATED WORK

There has been a recent surge in interest in compression in the deep neural network community. Denil et al. (2013) showed that by predicting parameters of neural networks there is great redundancy in the amount of parameters being used. This suggests that pruning, originally introduced to reduce structure in neural networks and hence improve generalization, can be applied to the problem of compression and speed-up (LeCun et al., 1989). In fact, (Han et al., 2015b; Guo et al., 2016) show that neural network survive severe weight pruning (up to 99%) without significant loss of accuracy. A variational version is is proposed by Molchanov et al. (2017), the authors learn the dropout rate for each weight in the network separately. Some parameters will effectively be pruned when the dropout rate is very high. In an approach slightly orthogonal to weight pruning, (Wen et al., 2016) applied structural regularization to prune entire sets of weights from the neural network. Such extreme weight pruning can lead to entire structures being obsolete, which for the case of convolutional filters, can greatly speed up prediction. Most importantly for compression, however, is that in conjunction with Compressed Sparse Column (CSC) format, weight pruning is a highly effective way to store and transfer weights. In Appendix A we discuss CSC format in more detail.

Reducing the bit size per stored weight is another approach to model compression. For example, reducing 32 bit floats to 1 bit leads to a $32\times$ storage improvement. Gong et al. (2014) proposed and experimented with a number of quantization approaches: binary quantization, k-means quantization, product quantization and residual quantization. Other work finds optimal fixed points (Lin et al., 2015), applies hashing (Chen et al., 2015) or minimizes the estimation error (Wu et al., 2015). Merolla et al. (2016) demonstrates that neural networks are robust against certain amounts of low precision; indeed several groups have exploited this and showed that decreasing the weight encoding precision has little to no effect on the accuracy loss (Gupta et al., 2015; Courbariaux et al., 2014; Venkatesh et al., 2016). Pushing the idea of extreme quantization, (Courbariaux et al., 2015) and Courbariaux & Bengio (2016) trained networks from scratch that use only 1bit weights with floating point gradients; to achieve competitive results, however, they require many more of these weights.

Han et al. (2015a) elaborate on combining these ideas. They introduce an multi-step algorithm that compresses CNNS up to $49\times$. First, weights are pruned (giving $9-13\times$ compression); second they quantize the weights (increasing compression to $27-31\times$); and last, they apply Huffman Encoding (giving a final compression of $35-49\times$). The quantization step is trainable in that after each weight is assigned to a cluster centroid, the centroids get trained with respect to the original loss function. Note that this approach has several restrictions: the number of weights set to zero is fixed after the pruning step, as is the assignment of a weight to a given cluster in the second step. Our approach overcomes all these restrictions.

A final approach to compressing information is to apply low rank matrix decomposition. First introduced by (Denton et al., 2014) and Jaderberg et al. (2014), and elaborated on by using low rank filters (Ioannou et al., 2015), low rank regularization (Tai et al., 2015) or combining low rank decomposition with sparsity (Liu et al., 2015).

## 4 METHOD

This section presents the procedure of network compression as applied in the experiment section. A summary can be found in Algorithm 1.

### 4.1 GENERAL SET-UP

We retrain pre-trained neural networks with soft weight-sharing and factorized Dirac posteriors. Hence we optimize

$$\mathcal{L}(\mathbf{w}, \{\mu_j, \sigma_j, \pi_j\}_{j=0}^J) = \mathcal{L}^E + \tau \mathcal{L}^C \qquad (6)$$

$$= -\log p(\mathbf{T}|\mathbf{X}, \mathbf{w}) - \tau \log p(\mathbf{w}, \{\mu_j, \sigma_j, \pi_j\}_{j=0}^J), \qquad (7)$$

via gradient descent, specifically using Adam (Kingma & Ba, 2014). The KL divergence reduces to the prior because the entropy term does not depend on any trainable parameters. Note that, similar to (Nowlan & Hinton, 1992) we weigh the log-prior contribution to the gradient by a factor of $\tau = 0.005$. In the process of retraining the weights, the variances, means, and mixing proportions of all but one component are learned. For one component, we fix $\mu_{j=0} = 0$ and $\pi_{j=0} = 0.999$. Alternatively we can train $\pi_{j=0}$ as well but restrict it by a Beta distribution hyper-prior. Our Gaussian MM prior is initialized with $2^4 + 1 = 17$ components. We initialize the learning rate for the weights and means, log-variances and log-mixing proportions separately. The weights should be trained with approximately the same learning rate used for pre-training. The remaining learning rates are set to $5 \cdot 10^{-4}$. Note that this is a very sensitive parameter. The Gaussian mixtures will collapse very fast as long as the error loss does not object. However if it collapses too fast weights might be left behind, thus it is important to set the learning rate such that the mixture does collapse too soon. If the learning rate is too small the mixture will converge too slowly. Another option to keep the mixture components from collapsing is to apply an Inverse-Gamma hyperprior on the mixture variances.

### 4.2 INITIALIZATION OF MIXTURE MODEL COMPONENTS

In principle, we follow the method proposed by Nowlan & Hinton (1992). We distribute the means of the 16 non-fixed components evenly over the range of the pre-trained weights. The variances will be initialized such that each Gaussian has significant probability mass in its region. A good orientation for setting a good initial variance is weight decay rate the original network has been trained on. The trainable mixing proportions are initialized evenly $\pi_j = (1 - \pi_{j=0})/J$. We also experimented with other approaches such as distributing the means such that each component assumes an equal amount of probability. We did not observe any significant improvement over the simpler initialization procedure.

### 4.3 POST-PROCESSING

After re-training we set each weight to the mean of the component that takes most responsibility for it i.e. we quantize the weights. Before quantizing, however, there might be redundant components

as explained in section 2. To eliminate those we follow Adhikari & Hollmén (2012) by computing the KL divergence between all components. For a KL divergence smaller than a threshold, we merge two components as follows

$$\pi_{\text{new}} = \pi_i + \pi_j, \qquad \mu_{\text{new}} = \frac{\pi_i\mu_i + \pi_j\mu_j}{\pi_i + \pi_j}, \qquad \sigma_{\text{new}}^2 = \frac{\pi_i\sigma_i^2 + \pi_j\sigma_j^2}{\pi_i + \pi_j} \qquad (8)$$

for two components with indices $i$ and $j$.

Finally, for practical compression we use the storage format used in Han et al. (2015a) (see Appendix A).

---

**Algorithm 1** *Soft weight-sharing for compression*, our proposed algorithm for neural network model compression. It is divided into two main steps: network re-training and post-processing.

---

**Require:** $\tau \leftarrow$ set the trade-off between error and complexity loss
**Require:** $\Theta \leftarrow$ set parameters for gradient decent scheme such as learning rate or momentum
**Require:** $\alpha, \beta \leftarrow$ set gamma hyper-prior parameter (optional)
 $\mathbf{w} \leftarrow$ initialize network weights with pre-trained network weights
 $\theta = \{\mu_j, \sigma_j, \pi_j\}_{j=1}^J \leftarrow$ initialize mixture parameters (see Sec. 4.2)
 **while** $\mathbf{w}, \theta$ not converged **do**
 $\mathbf{w}, \theta \leftarrow \nabla_{\mathbf{w},\theta}\mathcal{L}^E + \tau\mathcal{L}^C$ update $\mathbf{w}$ and $\theta$ with the gradient decent scheme of choice
 **end while**
 $\mathbf{w} \leftarrow \underset{\mu_k}{\text{argmax}}\dfrac{\pi_k\mathcal{N}(\mathbf{w}|\mu_k, \sigma_k)}{\sum \pi_j\mathcal{N}(\mathbf{w}|\mu_j, \sigma_j)}$ compute final weight by setting it to the mean that takes most responsibility (for details see Sec. 4.3)

---

# 5 MODELS

We test our compression procedure on two neural network models used in previous work we compare against in our experiments:

(a) **LeNet-300-100** an MNIST model described in LeCun et al. (1998). As no pre-trained model is available, we train our own, resulting in an error rate of $1.89\%$.

(b) **LeNet-5-Caffe** a modified version of the LeNet-5 MNIST model in LeCun et al. (1998). The model specification can be downloaded from the Caffe MNIST tutorial page [2]. As no pre-trained model is available, we train our own, resulting in an error rate of $0.88\%$.

(c) **ResNets** have been invented by He et al. (2015) and further developed by He et al. (2016) and Zagoruyko & Komodakis (2016). We choose a model version of the latter authors. In accordance with their notation, we choose a network with depth 16, width $k = 4$ and no dropout. This model has 2.7M parameters. In our experiments, we follow the authors by using only light augmentation, i.e., horizontal flips and random shifts by up to 4 pixels. Furthermore the data is normalized. The authors report error rates of 5.02% and 24.03% for CIFAR-10 and CIFAR-100 respectively. By reimplementing their model we trained models that achieve errors 6.48% and 28.23%.

# 6 EXPERIMENTS

## 6.1 INITIAL EXPERIMENT

First, we run our algorithm without any hyper-priors, an experiment on LeNet-300-100. In Figure 1 we visualise the original distribution over weights, the final distribution over weight and how each weight changed its position in the training process. After retraining, the distribution is sharply peaked around zero. Note that with our procedure the optimization process automatically determines how many weights per layer are pruned. Specifically in this experiment, 96% of the first layer (235K

---

[2]`https://github.com/BVLC/caffe/blob/master/examples/mnist/lenet_train_`
`test.prototxt`

parameter), 90% of the second (30K) and only 18% of the final layer (10K) are pruned. From observations of this and other experiments, we conclude that the amount of pruned weights depends mainly on the number of parameters in the layer rather than its position or type (convolutional or fully connected).

Evaluating the model reveals a compression rate of 64.2. The accuracy of the model does not drop significantly from 0.9811 to 0.9806. However, we do observe that the mixture components eventually collapse, i.e., the variances go to zero. This makes the prior inflexible and the optimization can easily get stuck because the prior is accumulating probability mass around the mixture means. For a weight, escaping from those high probability plateaus is impossible. This motivates the use hyper-priors such as an Inverse-Gamma prior on the variances to essentially lower bound them.

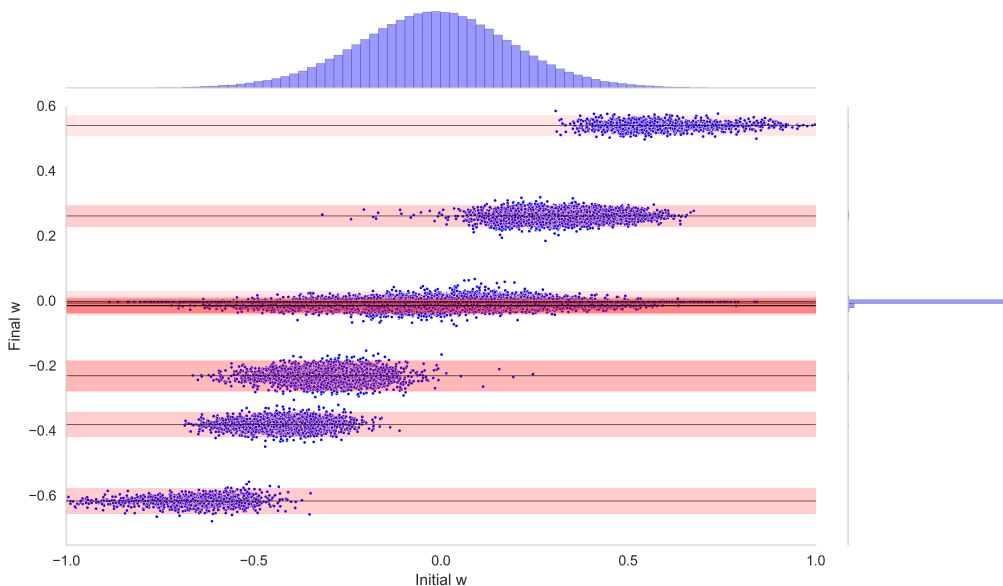

Figure 1: On top we show the distribution of a pretrained network. On the right the same distribution after retraining. The change in value of each weight is illustrated by a scatter plot.

## 6.2 HYPER-PARAMETER TUNING USING BAYESIAN OPTIMIZATION

The proposed procedure offers various freedoms: there are many hyper-parameters to optimize, one may use hyper-priors as motivated in the previous section or even go as far as using other distributions as mixture components.

To cope with the variety of choices, we optimize 13 hyper-parameters using the Bayesian optimization tool Spearmint Snoek et al. (2012). These include the learning rates of the weight and mixing components, the number of components, and $\tau$. Furthermore, we assume an Inverse-Gamma prior over the variances separately for the zero component and the other components and a Beta prior over the zero mixing components.

In these experiments, we optimize re-training hyperparameters for LeNet-300-100 and LeNet-5-Caffe. Due to computational restrictions, we set the number of training epochs to 40 (previously 100), knowing that this may lead to solutions that have not fully converged. Spearmint acts on an objective that balances accuracy loss vs compression rate. The accuracy loss in this case is measured over the training data. The results are shown in Figure 2. In the illustration we use the accuracy loss as given by the test data. The best results predicted by our spearmint objective are colored in dark blue. Note that we achieve competitive results in this experiment despite the restricted optimization time of 40 epochs, i.e. 18K updates.

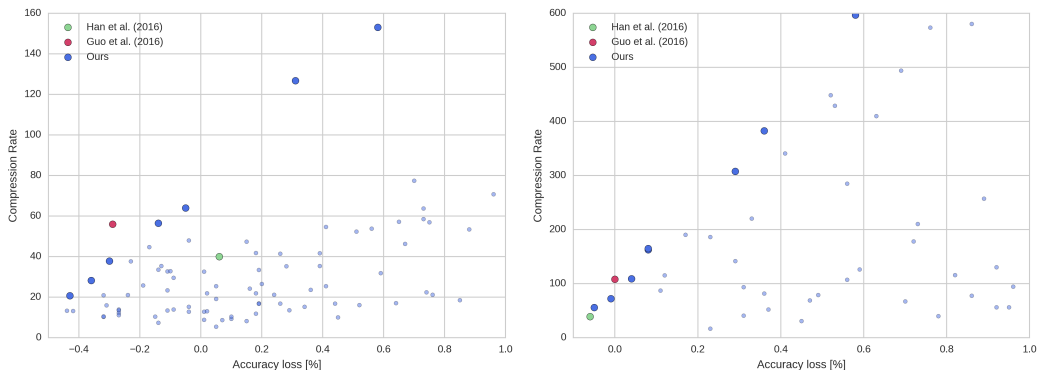

Figure 2: We show the results of optimizing hyper-parameters with spearmint. Specifically, we plot the accuracy loss of a re-trained network against the compression rate. Each point represents one hyper-parameter setting. The guesses of the optimizer improve over time. We also present the results of other methods for comparison. **Left**: LeNet-300-100 **Right**: LeNet-5-Caffe.

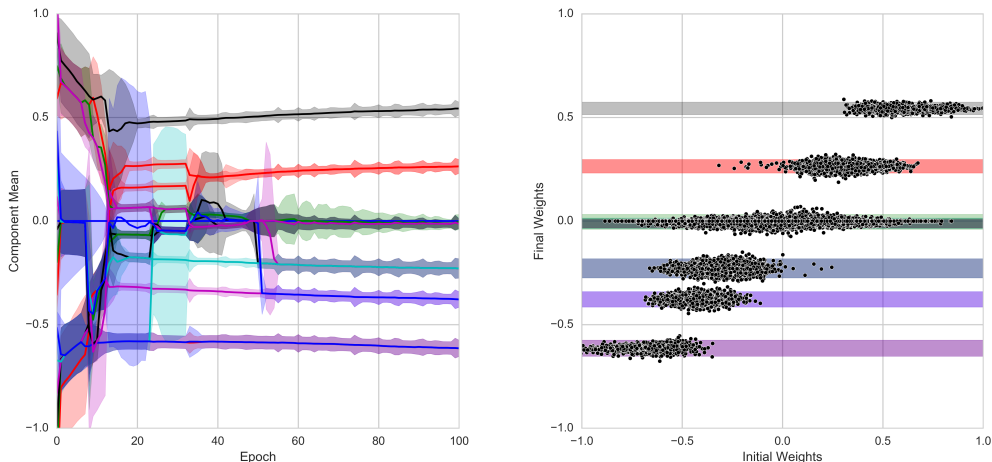

Figure 3: Illustration of our mixture model compression procedure on LeNet-5-Caffe. **Left**: Dynamics of Gaussian mixture components during the learning procedure. Initially there are 17 components, including the zero component. During learning components are absorbed into other components, resulting in roughly 6 significant components. **Right**: A scatter plot of initial versus final weights, along with the Gaussian components' uncertainties. The initial weight distribution is roughly one broad Gaussian, whereas the final weight distribution matches closely the final, learned prior which has become very peaked, resulting in good quantization properties.

The conclusions from this experiment are a bit unclear, on the one hand we do achieve state-of-the-art results for LeNet-5-Caffe, on the other hand there seems to be little connection between the parameter settings of best results. One wonders if a 13 dimensional parameter space can be searched efficiently with the amount of runs we were conducting. It may be more reasonable to get more inside in the optimization process and tune parameters according to those.

## 6.3 COMPRESSION RESULTS

We compare our compression scheme with Han et al. (2015a) and Guo et al. (2016) in Table 1. The results on MNIST networks are very promising. We achieve state-of-the-art compression rates in both examples. We can furthermore show results for a light version of ResNet with 2.7M parameters to illustrate that our method does scale to modern architectures. We used more components (64)

Table 1: **Compression Results**. We compare methods based on the post-processing error (we also indicate the starting error), the accuracy loss $\Delta$, the number of non zero weights $|\mathbf{W}_{\neq 0}|$ and the final compression rate CR based on the method proposed by Han et al. (2015a).

| Model | Method | Top-1 Error[%] | $\Delta$ [%] | $|\mathbf{W}|[10^6]$ | $\frac{|\mathbf{W}_{\neq 0}|}{|\mathbf{W}|}$ [%] | CR |
|---|---|---|---|---|---|---|
| LeNet-300-100 | Han et al. (2015a) | $1.64 \to 1.58$ | 0.06 | 0.2 | 8.0 | 40 |
| | Guo et al. (2016) | $2.28 \to 1.99$ | -0.29 | | 1.8 | 56 |
| | Ours | $1.89 \to 1.94$ | -0.05 | | 4.3 | **64** |
| LeNet-5-Caffe | Han et al. (2015a) | $0.80 \to 0.74$ | -0.06 | 0.4 | 8.0 | 39 |
| | Guo et al. (2016) | $0.91 \to 0.91$ | 0.00 | | 0.9 | 108 |
| | Ours | $0.88 \to 0.97$ | 0.09 | | 0.5 | **162** |
| ResNet (light) | Ours | $6.48 \to 8.50$ | 2.02 | 2.7 | 6.6 | 45 |

here to cover the large regime of weights. However, for large networks such as VGG with 138M parameters the algorithm is too slow to get usable results. We propose a solution for this problem in Appendix C; however, we do not have any experimental results yet.

## 7 DISCUSSION AND FUTURE WORK

In this work we revived a simple and principled regularization method based on soft weight-sharing and applied it directly to the problem of model compression. On the one hand we showed that we can optimize the MDL complexity lower bound, while on the other hand we showed that our method works well in practice when being applied to different models. A short-coming of the method at the moment is its computational cost and the ease of implementation. For the first, we provide a proposal that will be tested in future work. The latter is an open question at the moment. Note that our method—since it is optimizing the lower bound directly—will most likely also work when applied to other storage formats, such as those proposed originally by Hinton & Van Camp (1993). In the future we would like to extend beyond Dirac posteriors as done in Graves (2011) by extending the weight sharing prior to more general priors. For example, from a compression point of view, we could learn to prune entire structures from the network by placing Bernoulli priors over structures such as convolutional filters or ResNet units. Furthermore, it could be interesting to train models from scratch or in a student-teacher setting.

## ACKNOWLEDGEMENTS

We would like to thank Louis Smit, Christos Louizos, Thomas Kipf, Rianne van den Berg and Peter O'Connor for helpful discussions on the paper and the public code[3].

This research has been supported by Google.

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

# APPENDIX

## A  REVIEW OF STATE-OF-THE-ART NEURAL NETWORK COMPRESSION

We apply the compression scheme proposed by Han et al. (2015b;a) that highly optimizes the storage utilized by the weights. First of all, the authors store the weights in regular compressed sparse-row (CSR) format. Instead of storing $|W^{(l)}|$ parameters with a bit length of (commonly) $p_{\mathrm{orig}} = 32$ bit, CSR format stores three vectors (A, IR, IC).

- **A** stores all non-zero entries. It is thus of size $|W^{(l)}|_{\neq 0} \times p_{\mathrm{orig}}$, where $|W^{(l)}|_{\neq 0}$ is the number of non-zero entries in $W^{(l)}$.
- **IR** Is defined recursively: $\mathrm{IR}_0 = 0$, $\mathrm{IR}_k = \mathrm{IR}_{k-1} +$ (number of non-zero entries in the $(k-1)$-th row of $W^{(l)}$). It got $K + 1$ entries each of size $p_{\mathrm{orig}}$.
- **IC** contains the column index in $W^{(l)}$ of each element of A. The size is hence, $|W^{(l)}|_{\neq 0} \times p_{\mathrm{orig}}$.

An example shall illustrate the format, let

$$W^{(l)} = \begin{pmatrix} 0 & 0 & 0 & 1 \\ 0 & 2 & 0 & 0 \\ 0 & 0 & 0 & 0 \\ 2 & 5 & 0 & 0 \\ 0 & 0 & 0 & 1 \end{pmatrix}$$

than

$$\mathrm{A} = [1, 2, 2, 5, 1]$$
$$\mathrm{IR} = [0, 1, 2, 2, 4, 5]$$
$$\mathrm{IC} = [3, 1, 0, 1, 3]$$

The compression rate achieved by applying the CSC format naively is

$$r_p = \frac{|W^{(l)}|}{2|W^{(l)}|_{\neq 0} + (K + 1)} \tag{9}$$

However, this result can be significantly improved by optimizing each of the three arrays.

### A.1  STORING THE INDEX ARRAY IR

To optimize IR, note that the biggest number in IR is $|W^{(l)}|_{\neq 0}$. This number will be much smaller than $2^{p_{\mathrm{orig}}}$. Thus one could try to find $p \in \mathbf{Z}_+$ such that $|W^{(l)}|_{\neq 0} < 2^{p_{\mathrm{prun}}}$. A codebook would not be necessary. Thus instead of storing $(K + 1)$ values with $p_{\mathrm{orig}}$, we store them with $p_{\mathrm{prun}}$ depth.

### A.2  STORING THE INDEX ARRAY IC

Instead of storing the indexes, we store the differences between indexes. Thus there is a smaller range of values being used. We further shrink the range of utilized values by filling A with zeros whenever the distance between two non-zero weights extends the span of $2^p_{\mathrm{prun}}$. Han et al. (2015a) propose p = 5 for fully connected layers and p = 8 for convolutional layers. An illustration of the process can is shown in Fig. 4. Furthermore, the indexes will be compressed Hoffman encoding.

### A.3  STORING THE WEIGHT ARRAY A

In order to minimize the storage occupied by A. We quantize the values of A. Storing indexes in A and a consecutive codebook. Indexing can be improved further by again applying Huffman encoding.

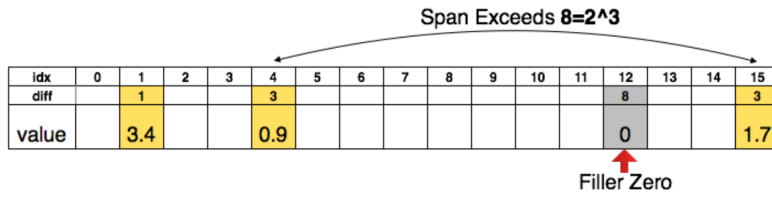

Figure 4: Illustration of the process described in A.2. IC is represented by relative indexes(diff). If the a relative index is larger than $8(=2^3)$, A will be filled with an additional zero. Figure from Han et al. (2015a).

# B CONFIGURING THE HYPER-PRIORS

## B.1 GAMMA DISTRIBUTION

The Gamma distribution is the conjugate prior for the precision of a univariate Gaussian distribution. It is defined for positive random variables $\lambda > 0$.

$$\Gamma(\lambda|\alpha, \beta) = \frac{\beta^\alpha}{\Gamma(\alpha)}\lambda^{\alpha-1}e^{-\beta\lambda} \tag{10}$$

For our purposes it is best characterised by its mode $\lambda^* = \dfrac{\alpha - 1}{\beta}$ and its variance $\text{var}_\gamma = \dfrac{\alpha}{\beta^2}$. In our experiments we set the desired variance of the mixture components to $0.05$. This corresponds to $\lambda^* = 1/(0.05)^2 = 400$. We show the effect of different choices for the variance of the Gamma distribution in Figure 5.

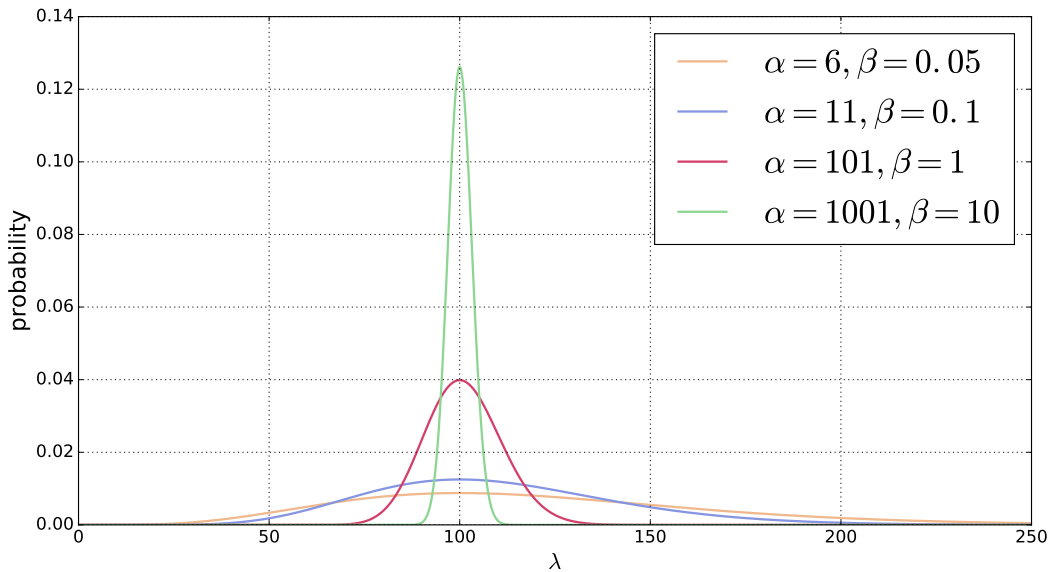

Figure 5: Gamma distribution with $\lambda^* = 100$. $\alpha$ and $\beta$ correspond to different choices for the variance of the distribution.

### B.2 Beta Distribution

The Beta distribution is the conjugate prior for the Bernoulli distribution, thus is often used to represent the probability for a binary event. It is defined for some random variable $\pi_{j=0} \in [0, 1]$

$$\mathcal{B}(\pi_{j=0}|\alpha, \beta) = \frac{\Gamma(\alpha + \beta)}{\Gamma(\alpha)\Gamma(\beta)}(\pi_{j=0})^{\alpha-1}(1 - \pi_{j=0})^{\beta-1} \tag{11}$$

with $\alpha, \beta > 0$. $\alpha$ and $\beta$ can be interpreted as the effective number of observations prior to an experiment, of $\pi_{j=0} = 1$ and $\pi_{j=0} = 0$, respectively. In the literature, $\alpha + \beta$ is defined as the pseudo-count. The higher the pseudo-count the stronger the prior. In Figure 6, we show the Beta distribution at constant mode $\pi_{j=0}^* = \frac{\alpha - 1}{\alpha + \beta - 2} = 0.9$. Note, that, the beta distribution is a special case of the Dirichlet distribution in a different problem setting it might be better to rely on this distribution to control all $\pi_j$.

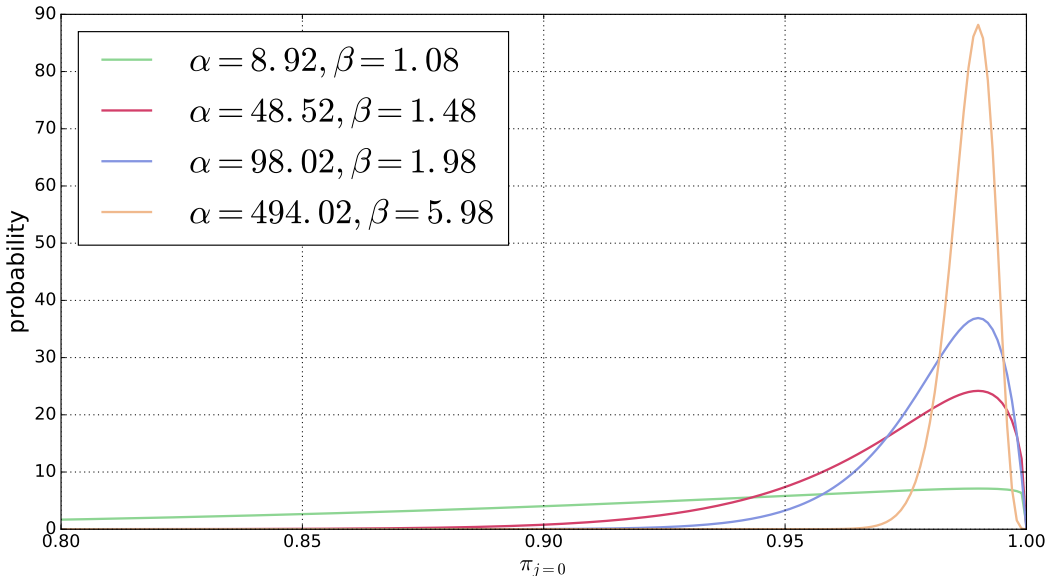

Figure 6: Beta distribution with $\pi_{j=0}^* = 0.9$. $\alpha$ and $\beta$ correspond to different choices for the pseudo-count.

## C    Scalability

Neural Networks are usually trained with a form of batch gradient decent (GD) algorithm. These methods fall into the umbrella of stochastic optimization (Robbins & Monro, 1951). Here the model parameters $\mathbf{W}$ are updated iteratively. At each iteration $t$, a set of $B$ data instances is used to compute a noisy approximation of the posterior derivative with respect to $\mathbf{W}$ given all data instances $N$.

$$\nabla_{\mathbf{W}} \log p(\mathbf{W}|\mathcal{D}) = \frac{N}{B} \sum_{n=1}^{B} \nabla_{\mathbf{W}} \log p(\mathbf{t}_n|\mathbf{x}_n, \mathbf{w}) + \sum_{i=1}^{I} \nabla_{\mathbf{W}} \log p(w_i) \tag{12}$$

This gradient approximation can subsequently be used in various update schemes such as simple GD.

For large models estimating the prior gradient can be an expensive operation. This is why we propose to apply similar measures for the gradient estimation of the prior as we did for the likelihood term. To do so, we sample K weights randomly. The noisy approximation of the posterior derivative is

now:

$$\nabla_{\mathbf{W}} \log p(\mathbf{W}|\mathcal{D}) = \frac{N}{B} \sum_{n=1}^{B} \nabla_{\mathbf{W}} \log p(\mathbf{t}_n|\mathbf{x}_n, \mathbf{w}) + \frac{I}{K} \sum_{i=1}^{K} \nabla_{\mathbf{W}} \log p(w_i) \qquad (13)$$

## D  FILTER VISUALISATION

In Figure D we show the pre-trained and compressed filters for the first and second layers of LeNet-5-Caffe. For some of the feature maps from layer 2 seem to be redundant hence the almost empty columns. In Figure D we show the pre-trained and compressed filters for the first and second layers of LeNet-300-100.

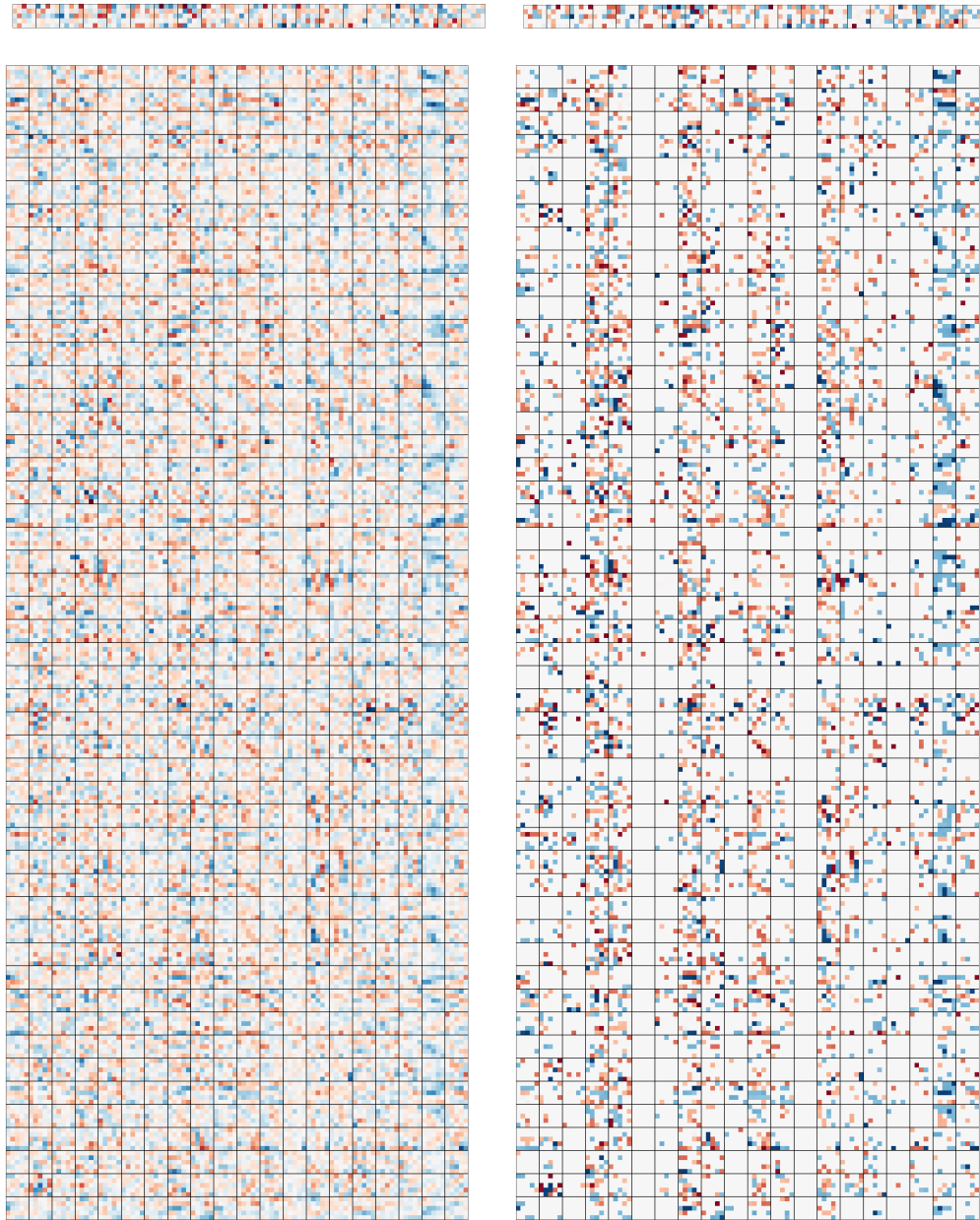

Figure 7:  Convolution filters from LeNet-5-Caffe. **Left:** Pre-trained filters. **Right:** Compressed filters. The top filters are the 20 first layer convolution weights; the bottom filters are the 20 by 50 convolution weights of the second layer.

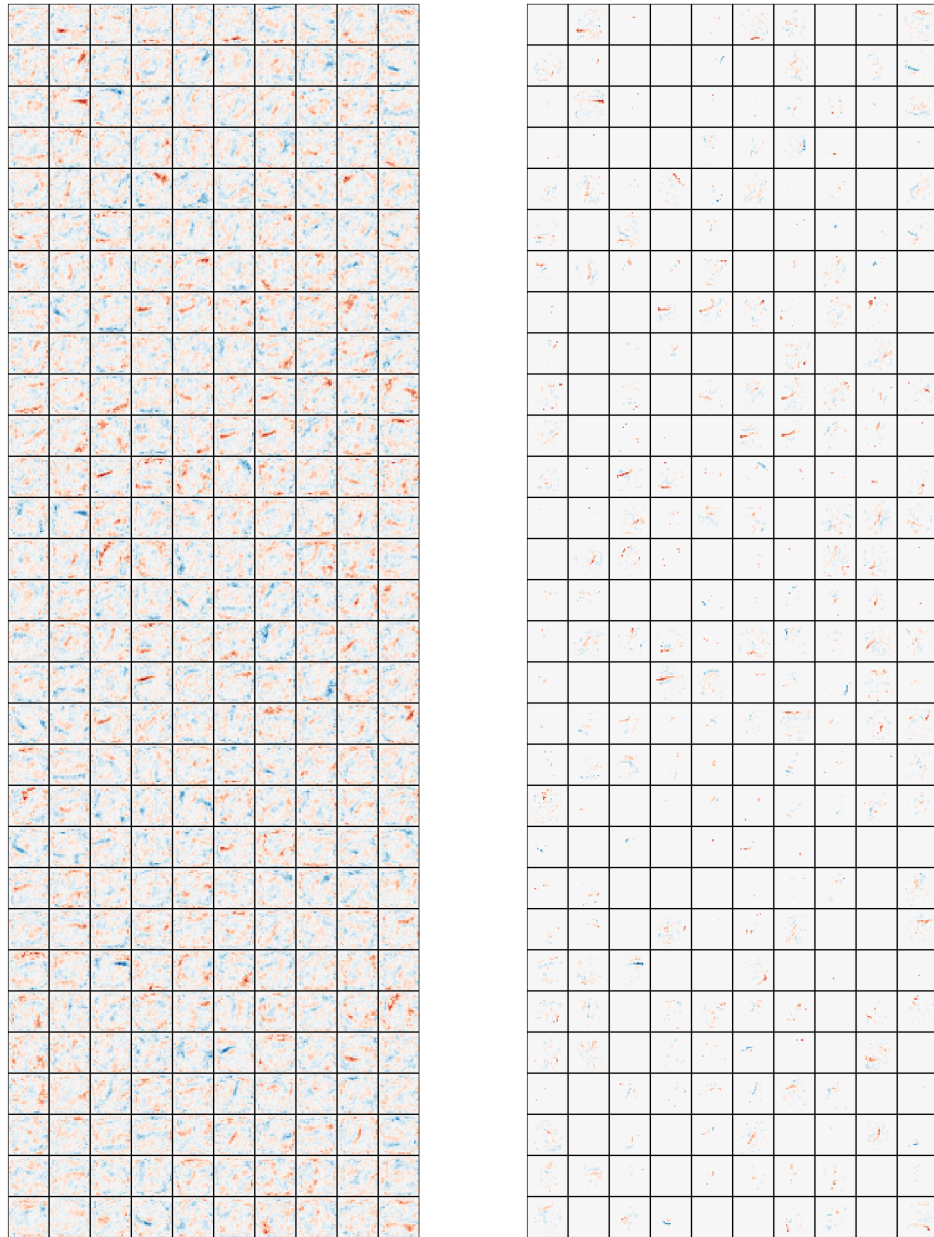

Figure 8: Feature filters for LeNet-300-100. **Left:** Pre-trained filters. **Right:** Compressed filters.

