# Peer review of "Soft Weight-Sharing for Neural Network Compression"

_ICLR 2017 — accepted_

[Author Response · Karen Ullrich · 13 Dec 2016]
**Paper Update**

We present an updated version of the paper:

(1)  We clarified the section that connects MDL,variational learning and compression and the methods section.

(2) We added the results from a spearmint experiment. 

(3) "We achieve state-of-the-art compression rates in both examples. However, for large networks such as VGG with 138 million parameters the algorithm as is, is too slow to get usable results. In experiences with VGG we were able to prune 93% of the weights without loss of accuracy, however, the quantization step resulted in significant loss of accuracy. We think this is due to the network not having convergened."

(4) Visualizations of the filters of the second convolutional layer of LeNet-5-Caffe.

[Official Review · AnonReviewer1 · rating 7 · confidence 3 · 16 Dec 2016]
**Empirical Bayesian learning applied for neural net parameter compression**
soundness 3 · clarity 4

This paper proposes to use an empirical Bayesian approach to learn the parameters of a neural network, and their priors.
A mixture model prior over the weights leads to a clustering effect in the weight posterior distributions (which are approximated with delta peaks). 
This clustering effect can exploited for parameter quantisation and compression of the network parameters.
The authors show that this leads to compression rates and predictive accuracy comparable to related approaches. 

Earlier work [Han et al. 2015] is based on a three-stage process of pruning small magnitude weights, clustering the remaining ones, and updating the cluster centres to optimise performance. The current work provides a more principled approach that does not have such an ad-hoc multi-stage structure, but a single iterative optimisation process.

A first experiment, described in section 6.1 shows that an empirical Bayes’ approach, without the use of hyper priors, already leads to a pronounced clustering effect and to setting many weights to zero. 
In particular a compression rate of 64.2 is obtained on the LeNet300-100 model.
In section 6.1 the text refers to figure C, I suppose this should be figure 1.

Section 6.2 describes an experiment where hyper-priors are used, and the parameters of these distributions, as well as other hyper-parameters such as the learning rates, are being optimised using Spearmint (Snoek et al., 2012). Figure 2 shows the performance of the  different points in the hyper-parameter space that have been evaluated (each trained network gives an accuracy-compressionrate point in the graph). The text claims that best results lie on a line, this seems a little opportunistic interpretation given the limited data. Moreover, it would be useful to add a small discussion on whether such a linear relationship would be expected or not. Currently the results of this experiment lack interpretation.

Section 6.3 describes results obtained for both CNN models and compares results to the recent results of (Han et al., 2015) and (Guo et al., 2016).
Comparable results are obtained in terms of compression rate and accuracy. 
The authors state that their current algorithm is too slow to be useful for larger models such as VGG-19, but they do briefly report some results obtained for this model (but do not compare to related work). It would be useful here to explain what slows the training down with respect to standard training without the weight clustering approach, and how the proposed algorithm scales in terms of the relevant quantities of the data and the model.

The contribution of this paper is mostly experimental, leveraging fairly standard ideas from empirical Bayesian learning to introduce weight clustering effects in CNN training.
This being said, it is an interesting result that such a relatively straightforward approach leads to results that are on par with state-of-the-art, but more ad-hoc, network compression techniques.
The paper could be improved by clearly describing the algorithm used for training, and how it scales to large networks and datasets.
Another point that would deserve further discussion is how the hyper-parameter search is performed ( not using test data I assume), and how the compared methods dealt with the search over hyper-parameters to determine the accuracy-compression tradeoff. Ideally, I think, methods should be evaluated across different points on this trade-off.

[Official Review · AnonReviewer3 · rating 7 · confidence 4 · 16 Dec 2016]
**Nice idea, some minor issues**
soundness 3

The authors propose a method to compress neural networks by retraining them while putting a mixture of Gaussians prior on the weights with learned means and variances which then can be used to compress the neural network by first setting all weights to the mean of their infered mixture component (resulting in a possible loss of precision) and storing the network in a format which saves only the fixture index and exploits the sparseness of the weights that was enforced in training.

Quality:
Of course it is a serious drawback that the method doesn't seem to work on VGG which would render the method unusable for production (as it is right now, maybe this can be improved). I guess AlexNet takes too long to process, too, otherwise this might be a very valuable addition.
In Figure 2 I am noticing two things: On the left, there is a large number of points with improved accuracy which is not the case for LeNet5-Caffe. Is there any intuition for why that's the case? Additionally regarding the spearmint optimization: Do they authors have found any clues about which hyperparameter settings worked well? This might be helpful for other people trying to apply this method.
I really like Figure 7 in it's latest version.

Clarity:
Especially section 2 on MDL is written very well and gives a nice theoretic introduction. Sections 4, 5 and 6 are very short but seem to contain most relevant information. It might be helpful to have at least some more details about the used models in the paper (maybe the number of layers and the number of parameters).
In 6.1 the authors claim "Even though most variances seem to be reasonable small there are some that are large". From figure 1 this is very hard to assess, especially as the vertical histogram essentially shows only the zero component. It might be helpful to have either a log histogram or separate histograms for each componenent. What are the large points in Figure 2 as opposed to the smaller ones? They seem to have a very good compression/accuracy loss ratio, is that it?
Some other points are listed below

originality: While there has been some work on compressing neural networks by using a reduced number of bits to store the parameters and exploiting sparsity structure, I like the idea to directly learn the quantization by means of a gaussian mixture prior in retraining which seems to be more principled than other approaches

significance: The method achievs state-of-the-art performance on the two shown examples on MNIST, however these networks are far from the deep networks used in state-of-the-art models. This obviously is a drawback for the practical usability of the methods and therefor it's significance. If the method could be made to work on more state-of-the-art networks like VGG or ResNet, I would consider this a contribution of high significance.

Minor issues:

page 1: There seems to be a space in front of the first author's name
page 3: "in this scenario, pi_0 may be fixed...". Missing backslash in TeX?
page 6: 6.2: two wrong blanks in "the number of components_, \tau_."
page 6, 6.3: "in experiences with VGG": In experiments?
page 12: "Figure C": Figure 7?

[Official Review · AnonReviewer4 · rating 7 · confidence 3 · 16 Dec 2016]
**nice tie-in to classic NN literature through lens of modern engineering needs**

This paper revives a classic idea involving regularization for purposes of compression for modern CNN models on resource constrained devices. Model compression is hot and we're in the midst of lots of people rediscovering old ideas in this area so it is nice to have a paper that explicitly draws upon classic approaches from the early 90s to obtain competitive results on standard benchmarks.

There's not too much to say here: this study is an instance of a simple idea applied effectively to an important problem, written up in an illuminating manner with appropriate references to classic approaches. The addition of the filter visualizations enhances the contribution.

[Author Response · Karen Ullrich · 24 Dec 2016]
**Comment on reviews**

Thank you very much for the carefully crafted reviews. We agree with most points you are making. We would like to comment on them and give an outlook of future efforts.
 
VGG: We ran our algorithm that was successful for MNIST on ImageNet using the pretrained VGG network. However, due to some implementation inefficiencies we were only able to run it for a few epochs. Therefore, we have strong reasons to believe the network has not converged the way that LeNet had. In order to quickly verify that our method works on natural images we ran it on CIFAR10/100 and the results look in line with our LeNet experiments. Our computational bottleneck seems to stem from the updating of the prior parameters. In the future, we will be updating prior parameters with influence of less weights.  

Clarity: The methods and experiments section will be improved and extended to enhance its clarity. In particular, we will add an explicit algorithm. Furthermore, we will put our code up online.

Spearmint: We agree with AnonReviewer1 that there is no theoretical justification for a linear relationship of accuracy and compression rate. We can derive some theoretical results under the assumption that the amount of pruned weights and the accuracy have a known relationship and considering only the storage format proposed by Han et al. (2016).
We will improve that in the next upcoming version.

In response to: "In Figure 2 I am noticing two things: On the left, there is a large number of points with improved accuracy which is not the case for LeNet5-Caffe. Is there any intuition for why that's the case?"
Nowlan and Hinton (1992) did originally propose the method to improve generalization. They offered it as an alternative to convolutional weight-sharing. I think for this fully connected architecture that is exactly what is happening. The network has more room for generalization because there has been no other form of regularization.

We hope to be able to report more soon ... next year :).

Till than happy Christmas and New Year.

[Author Response · Karen Ullrich · 19 Jan 2017]
**Final update and comments**

Dear reviewers,


We would like to announce our final update, main changes include:

(A) A Pseudo-Algorithm for more clarity of the process

(B) Clarified description of the method and experiment section in particular.

(C) Results for wide ResNets on CIFAR-10.

(D) Proposal for prior update given extremely many parameters such as VGG in Appendix C.



We also would like to make some final comments on the procedure.

We do believe that that our method is principled and will achieve high compression rates even when the format of storing changes, because we optimize the lower bound directly. One could imagine a scenario where we store noisy weights as proposed by Hinton et al. (1992).

However, there is to say that the process involves a lot of hyper parameters that are hard to tune.
Interesting, also, that there seems to be a regime where the empirical prior helps improving upon pretrained results (often very significantly). For us that encourages using empirical priors for neural network training in general such as training from scratch, training in a teacher student setting or training networks with little data.

[Author Response · Karen Ullrich · 15 Feb 2017]
**UPDATE: code now available**

Along the paper we publish a little tutorial. It contains the basic functionalities.

[Final Decision · Program Chairs · 06 Feb 2017]
**ICLR committee final decision**

This paper provides a principled and practical formulation for weight-sharing and quantization, using a simple mixture of Guassians on the weights, and stochastic variational inference. The main idea and results are presented clearly, along with illustrative side-experiments showing the properties of this method in practice. Also, the method is illustrated on non-toy problems.